# Diabetic kidney disease in northwest Ethiopia: Prevalence and determinants among adults with type 2 diabetes

Workagegnehu Hailu[1]*, Tadesse Asmamaw Dejene[2], Markeshaw Tiruneh[2], Meseret Derbew Molla[2], Eshetie Melese Birru[3,4,5], Shitaye Alemu[1], Tadesse Awoke[6]

1 Department of Internal Medicine, College of Medicine and Health Sciences, University of Gondar, Gondar, Ethiopia, 2 Department of Medical Biochemistry, College of Medicine and Health Sciences, University of Gondar, Gondar, Ethiopia, 3 Department of Pharmacology, College of Medicine and Health Sciences, University of Gondar, Gondar, Ethiopia, 4 The Kids Research Institute Australia, Nedlands, Western Australia, Australia, 5 Medical School, the Uiniversity of Western Australia, Crawley, Western Australia, Australia, 6 Department Epidemiology and Biostatistics, College of Medicine and Health Sciences, University of Gondar, Gondar, Ethiopia

* workhailu@yahoo.com

## Abstract

### Background

Diabetic kidney disease (DKD), mainly due to type 2 diabetes (T2DM) is the leading cause of end-stage kidney disease globally. However, DKD prevalence in sub-Saharan Africa, particularly Ethiopia, is underexplored, especially using reliable markers like quantified albuminuria and cystatin C based estimated glomerular equations (eGFR). This study aimed to assess DKD prevalence and associated factors using multiple diagnostic markers.

### Methods

A cross-sectional study was conducted in adult T2DM patients at the University of Gondar Comprehensive Specialized Hospital using systematic random sampling. Data on socio-demographics and lab parameters were collected, with DKD diagnosed via eGFR and/or albuminuria (spot urine albumin-to-creatinine ratio and 24-hour collection). SPSS version 28 was used for data analysis, and factors were identified through multivariable logistic regression, with significance at 95% CI and p < 0.05.

### Results

In a study of 204 T2DM patients (mean age 60.2 years; 57.4% female), the prevalence of DKD was 37.3% (95% CI: 30.6–44.3). Significant factors associated with DKD included urban residence (AOR = 0.278, p = 0.023), poor blood pressure control (AOR = 2.33, p = 0.016), poor glycemic control (AOR = 2.93, p = 0.007), and longer diabetes duration (AOR = 6.78, p < 0.0001).

**Data availability statement:** All relevant data are within the manuscript and its Supporting information files.

**Funding:** Funding for this research was obtained from the University of Gondar, College of Medicine and Health Sciences- research, technology transfer, and community engagement. (Ref No. R/T/T/C/Eng./189/11/22). The funders had no role in study design, data collection and analysis, decision to publish, or preparation of the manuscript.

**Competing interests:** The authors have declared that no competing interests exist.

**Abbreviations:** ACR, Albumin-to-Creatinine Ratio; AER, Albumin Excretion Rate; AKI, Acute Kidney Injury; BMI, Body Mass Index; CKD, Chronic Kidney Disease; CKD-EPI, Chronic Kidney Disease Epidemiology Collaboration; CI, Confidence Interval; COR, Crude Odds Ratio; DKD, Diabetic Kidney Disease; DN, Diabetic Nephropathy; eGFR, Estimated Glomerular Filtration Rate; ESKD, End-Stage Kidney Disease; KIDIGO, Kidney Disease: Improving Global Outcomes; LMICs, Low- and Middle-Income Countries; MTF, Metformin; SD, Standard Deviation; SU, Sulfonylurea; T2DM, Type 2 Diabetes Mellitus.

## Conclusion

This study shows a high prevalence of DKD in T2DM patients, mainly identified via albuminuria. Poor blood pressure control, inadequate glycemic control, and longer diabetes duration were significantly associated with DKD. Regular screening and improved glycemic and blood pressure control are essential to slow DKD progression.

## Introduction

Diabetes mellitus is a metabolic disorder characterized by chronic hyperglycemia due to defects in insulin secretion, action, or both [1]. The International Diabetes Federation estimates that 537 million adults globally have diabetes, with projections reaching 783 million by 2045; over 75% of these individuals live in low- and middle-income countries (LMICs). In Ethiopia, the prevalence of diabetes among adults is estimated at 3.3%. [2]. Type 2 diabetes mellitus (T2DM) accounts for 87% to 91% of the global diabetes burden [3,4].

Chronic kidney disease (CKD) is defined as abnormalities in kidney structure or function lasting over three months that have health implications. it is classified based on the cause, GFR category (G1–G5), and albuminuria category (A1–A3) [5]. Nearly half of patients with T2DM develop diabetic kidney diseases(DKD)during their lifetime. Globally, DKD is the leading cause of CKD and end-stage kidney disease (ESKD), accounting for 50% of cases in developed countries [4,6].

DKD is defined as persistently elevated urine albumin excretion (albumin-to-creatinine ratio [ACR] ≥ 30 mg/g) and/or a reduced estimated glomerular filtration rate (eGFR < 60 mL/min/1.73 m²) in individuals with diabetes mellitus, lasting for at least three months [5]. While DKD typically progresses from microalbuminuria to macroalbuminuria, there are diverse clinical presentations, with some patients experiencing renal function decline without albuminuria [7,8].

The prevalence of DKD is increasing, particularly in LMICs, yet it remains under-recognized and often diagnosed late [9].The reported prevalence of DKD varies across studies and this may arise from regional differences, methods of GFR estimation, variations in albuminuria measurement, as well as racial differences and disparities in economic resources and healthcare access [10,11]. Significant disparities exist in the prevalence of DKD among the major ethnic groups [12,13].

Evidence on kidney disease burden among diabetics in Africa is limited. A systematic review indicated that urine protein measurement was the most common method for assessing kidney damage, with CKD prevalence in Africa ranging from 11% to 83.7% [14]. In sub-Saharan Africa, the prevalence of microalbuminuria among T2DM patients was found to be 40.24% [15]. Similarly, another systematic review and meta-analysis in sub-Saharan Africa reported an overall pooled prevalence of DKD at 35.3% with 29.7% in Eastern Africa [16].

In Ethiopia, systematic reviews show CKD prevalence among diabetes patients ranging from 18.22% to 35.5% [17,18]. Institution based studies in Ethiopia including in our study area reposted prevalence of CKD ranging 17.3% to 26% [19–24].

Individual studies [19–24] in Ethiopia exhibit considerable variability in methodology and definitions. Many of these studies rely on creatinine and eGFR without assessing albuminuria or use unreliable tests for albuminuria, such as urine dipstick tests. This may underestimate the true prevalence. Newer GFR estimation methods, such as cystatin C-based formulas, offer advantages for early identification of declining renal function [25]. Consequently, the true prevalence of DKD in Ethiopia, using more reliable diagnostic markers such as quantified albuminuria and eGFR based on various markers, including serum cystatin C, remains inadequately addressed.

Therefore, the aim of this study was to comprehensively assess the prevalence of DKD in patients with T2DM by using different diagnostic markers, including 24-hours urine albumin, spot albumin-to-creatinine ratio, and eGFR using both serum creatinine and cystatin C measurements. This approach aimed to fill the existing gaps in the literature and provide a more accurate understanding of DKD prevalence in the area.

## Methods

### Study design and setting

This cross-sectional, institution-based study was conducted at the University of Gondar Comprehensive Specialized Hospital (UoGCSH), Gondar, Ethiopia, from October 1, 2023 to January 31, 2024. The hospital is located in the historic town of Gondar, approximately 750 km from the capital, Addis Ababa. It is one of Ethiopia's oldest and most pioneering healthcare institutions, with a bed capacity of 800, providing both teaching and referral services to a catchment area of over 7 million people in the region and neighboring areas.

The Department of Internal Medicine manages various diseases, including diabetes mellitus, hypertension, CKD, and other cardiovascular conditions. The hospital's diabetes clinic is well-established and operates daily, serving an average of 50 diabetic patients daily. Additionally, there are regular biweekly nephrology and hypertension clinics, where approximately 25 patients with kidney disease and 80 patients with hypertension patients are seen weekly.

The hospital's dialysis unit, equipped with four functional hemodialysis machines, provides hemodialysis services for patients with acute kidney injury (AKI) and kidney failure. However, the dialysis service is only available to those who can afford to pay out of pocket. There is no kidney transplant service available.

Basic laboratory tests, such as serum glucose, urea, creatinine, and urine dipstick tests, are routinely available at the hospital. However, more specialized tests like hemoglobin A1c, quantitative urine protein measurements (such as the albumin-to-creatinine ratio and 24-hour urine protein), and serum cystatin C measurements are unavailable. This study follows the Strengthening the Reporting of Observational Studies in Epidemiology (STROBE) guidelines for the design, analysis, and reporting to ensure transparent and accurate documentation (https://www.strobe-statement.org/). A completed STROBE checklist is provided in Table S1 in S1 File.

### Source and study population

The source population for this study consisted of all patients with T2DM receiving follow-up care at UoGCSH. The study participants were both male and female adult patients with T2DM who attended the diabetes follow-up clinics during the study period. All adults aged 18 years and above with an established diagnosis of T2DM and currently taking antidiabetic medications were included. Individuals who were critically ill or diagnosed with Type 1 Diabetes Mellitus were excluded.

### Sample size and sampling technique

The sample size was determined based on previous reports of 17.3% prevalence of CKD among patients with T2DM [22]. Using a 95% confidence level, a precision of 0.5, and accounting for a 10% contingency, the sample size was calculated using a single proportion formula and determined to be 200. Systematic random sampling was used to select participants with T2DM.

## Variables and operational definitions

The outcome variable of interest was the presence of diabetic kidney disease (DKD). The independent variables included demographic factors such as age and sex, duration of diabetes, diabetes control status, and comorbidities.

DKD was defined as elevated urine albumin excretion (albumin-to-creatinine ratio [ACR] ≥ 30 mg/g or 24-hour albumin excretion rate [AER] > 30 mg/day) and/or a reduced estimated glomerular filtration rate (eGFR < 60 mL/min/1.73 m²) in individuals with diabetes mellitus [5]. For the purpose of this study, we used the 24-hour AER, as it is the most precise test to measure albuminuria. To estimate eGFR, we applied the CKD-EPI equation, which utilizes both serum creatinine and cystatin C levels.

CKD and albuminuria definition and classification were according to the Kidney Disease Improving Global Outcomes (KDIGO) guideline: GFR category, and albuminuria category [26].

The GFR categories are defined as follows: G1 indicates a GFR of ≥ 90 mL/minute/1.73 m²; G2 represents a GFR of 60–89 mL/minute/1.73 m²; G3a corresponds to a GFR of 45–59 mL/minute/1.73 m²; G3b indicates a GFR of 30–44 mL/minute/1.73 m²; G4 is defined as a GFR of 15–29 mL/minute/1.73 m²; and G5 reflects a GFR of < 15 mL/minute/1.73 m². Regarding albuminuria, A1 signifies urinary AER < 30 mg/24 hours and UACR < 30 mg/g (normal-to-mildly increased), A2 indicates urinary AER of 30–299 mg/24 hours and UACR of 30–299 mg/g (moderately increased), while A3 corresponds to urinary AER ≥ 300 mg/24 hours and UACR ≥ 300 mg/g (severely increased).

"Controlled blood pressure" was defined as blood pressure < 140/90 mmHg while on antihypertensive treatment, whereas "uncontrolled blood pressure" was defined as blood pressure ≥ 140/90 mmHg while on antihypertensive treatment.

"Controlled diabetes mellitus" was defined as hemoglobin A1c (A1c) < 7% while on antidiabetic treatment, while "uncontrolled diabetes mellitus" was defined as A1c ≥ 7% while on treatment.

Current smokers were defined as those who reported smoking any number of cigarettes every day or some days

Alcohol consumption is considered to be "positive" if an individual has consumed any amount of alcohol within the past 12 months and 'negative' if no alcohol intake in the past 12 months.

## Data collection methods and procedures

Data were collected through a pre-tested, interviewer-administered structured questionnaire. Information on demographic factors, including age, sex, place of residence, education level, occupation, alcohol consumption, smoking behavior, and physical activity was obtained by interviewing the participants. Additional clinical and laboratory findings, such as the duration of diabetes mellitus, comorbidities, and type of medication used (oral hypoglycemic agents, insulin, or both), were collected from individual medical record reviews.

Height and weight were measured to calculate body mass index (BMI), which was then classified according to WHO categories. Blood pressure was measured using an appropriate cuff size and an aneroid sphygmomanometer to assess hypertension and control status.

## Blood and urine sample collection and processing

Blood and urine samples were collected from each participant by trained laboratory technologists. Blood samples were obtained through venipuncture, and biochemical analytes such as cystatin C and creatinine were measured in the serum.

A first-morning midstream urine sample was collected to measure a spot urine albumin-to-creatinine ratio (ACR). The ACR measurement was performed once and was not repeated. Urine ACR was measured using the Kinetic Alkaline Picrate (Jaffe Reaction) method on the Abbott Alinity instrument (Abbott Laboratories, USA). Additionally, 24-hours urine samples were collected from all participants after providing with appropriate verbal and written instructions on collection methods. These samples were used to measure 24-hours urine albumin levels using the COBAS 6000 analyzer (Roche, Switzerland).

 

Hemoglobin A1C was measured from whole blood collected with EDTA using an enzymatic assay on the Abbott Alinity instrument. Serum creatinine and cystatin C were measured using the automated particle-enhanced immunoturbidimetric method on the COBAS 6000 analyzer, based on the enzymatic colorimetric principle. The values of serum creatinine and cystatin C were then used to estimate GFR using the race-free 2021 CKD-EPI equation [27]. Three eGFR calculations were performed for each patient: eGFRcr (based on creatinine alone), eGFRcys (based on cystatin C alone), and eGFRcr-cys (based on both creatinine and cystatin C).

### Data quality assurance

To ensure data quality, the data collection tool was prepared in English and translated into the local language, Amharic, and then back-translated into English by the project team members. The questionnaire was pretested, and training was provided for the data collectors and supervisors. Data were collected by two MSc students in Biochemistry and supervised by a biochemist, a lab technologist, and the principal investigator (PI), who is a nephrologist.

### Statistical analysis

Data were checked, cleaned, coded, and entered using double entry into EpiData version 4.6 and exported to SPSS version 28 for statistical analysis. Continuous variables were expressed as mean ± standard deviation (SD), while categorical variables were presented as frequencies and percentages in tables and graphs. Descriptive statistics summarized demographic and clinical data. Bivariate and multivariate binary logistic regression analyses were performed to identify factors associated with DKD in T2DM patients. Variables with a P-value of less than 0.2 in the bivariate analysis were included in the multivariate analysis to control the effect of confounders. Associations were reported using adjusted odds ratios (AOR) with 95% confidence intervals (CI), and a P-value of less than 0.05 was considered statistically significant.

### Ethics approval and consent to participate

Ethical approval was obtained from the University of Gondar, college of medicine and health sciences, college ethics review committee (Ref No. R/T/T/C/Eng./187/11/22). Written informed consent was obtained from each participant.

## Results

### Socio demographic characteristics

A total of 210 T2DM patients were approached for participation in the study, and 204 provided complete responses, resulting in a response rate of 97.1%. The mean age of patients was 60.23 years (SD ± 10.66). More than half of the participants, 111 (54.41%), were under the age of 60, and with the majority being female 117(57.35%) (Table 1).

### Clinical and treatment related characteristics

The majority of participants, 128 (62.75%), had at least one comorbidity, with hypertension (83 participants, 63.36%) and dyslipidemia (35 participants, 26.72%) being the most common comorbidities. The mean duration of diabetes mellitus was 7.74 years (SD ± 5.79), while the median duration was 6.50 years (IQR: 3.00–10.75). Nearly three-quarters of the participants (72.06%) had poor diabetic control (Table 2).

### Albuminuria among patients with T2DM

Using a spot urine albumin-to-creatinine ratio test, nearly half (48.5%) of the patients had albuminuria (≥ 30 mg/g). In contrast, using a 24-hour urine albumin excretion rate, 33.3% of patients had albuminuria. The majority of patients fell within the range of moderately elevated albuminuria (Fig 1).

**Table 1. Socio demographic and behavioral characteristics of T2DM patients at University of Gondar Specialized Hospital. (N = 204), Gondar, Ethiopia, 2024.**

| Variables | Number | Percent |
|---|---|---|
| **Age, Mean(±SD)= 60.23(±10.66)** | | |
| **Sex of Patients** | | |
| Male | 87 | 42.65 |
| Female | 117 | 57.35 |
| **Age of patients (years)** | | |
| ≥60 | 93 | 45.59 |
| <60 | 111 | 54.41 |
| **Residence** | | |
| Urban | 184 | 90.20 |
| Rural | 20 | 9.80 |
| **Educational status** | | |
| Uneducated | 83 | 40.69 |
| Primary | 32 | 15.69 |
| High school | 48 | 23.53 |
| College and above | 41 | 20.10 |
| **Alcohol consumption** | | |
| Yes | 38 | 18.63 |
| No | 166 | 81.37 |
| **Smoking status** | | |
| Smoker | 4 | 1.86 |
| Non-smoker | 200 | 98.04 |
| **Occupational status** | | |
| Unemployed | 144 | 70.59 |
| Employed | 60 | 29.41 |
| **Marital status** | | |
| Single | 68 | 33.33 |
| Married | 136 | 66.67 |
| **Body mass index(BMI)** | | |
| Underweight | 4 | 1.96 |
| Normal | 104 | 50.98 |
| Overweight | 67 | 32.84 |
| Obesity | 29 | 14.22 |

## Prevalence of DKD and associated factors among patients with T2DM

The prevalence of DKD (defined by a 24-hour AER of ≥30 mg/24 hr or eGFR using cystatin C and creatinine of <60 mL/min/1.73 m², or both) was 37.3% (95% CI: 30.6–44.3) (Fig 2).

The proportions of albuminuria and reduced eGFR using different diagnostic markers are depicted in Table 3.

## Factors associated with DKD in patients with T2DM

A bivariate logistic regression was conducted for variables including age, gender, BMI, and the presence of comorbidities such as hypertension, cardiovascular disease, and dyslipidemia, as well as duration of diabetes, blood pressure (BP), and glycemic control status. Variables with a p-value of 0.20 or less—namely, place of residence, presence of comorbidities, BP control status, glycemic control status, and duration of diabetes—were selected for multivariable logistic regression.

**Table 2. Clinical and treatment related characteristics of type T2DM patients at University of Gondar Specialized Hospital. (N = 204), Gondar, Ethiopia, 2024.**

| Variables | Number | Percentage |
|---|---|---|
| **Comorbidity** | | |
| Yes | 128 | 62.75 |
| No | 76 | 37.25 |
| **Type of comorbidity** | | |
| Hypertension | 83 | 63.36 |
| Dyslipidemia | 35 | 26.72 |
| Others* | 13 | 9.92 |
| **Duration of diabetes** | | |
| <5 yrs | 65 | 31.86 |
| ≥5 yrs | 139 | 68.14 |
| **Glycemic control status** | | |
| Good control | 57 | 27.94 |
| Poor control | 147 | 72.06 |
| **Type of anti-diabetic treatment** | | |
| Metformin(MTF)alone | 67 | 32.84 |
| Sulfonylureas (SU)alone | 4 | 1.96 |
| MTF + SU | 71 | 34.80 |
| Insulin alone | 29 | 14.22 |
| Insulin + MTF | 33 | 16.18 |
| **Blood Pressure control status** | | |
| Controlled | 163 | 79.90 |
| Uncontrolled | 41 | 20.10 |

• Stroke, ischemic heart disease, Bronchial asthma, Gout arthritis.

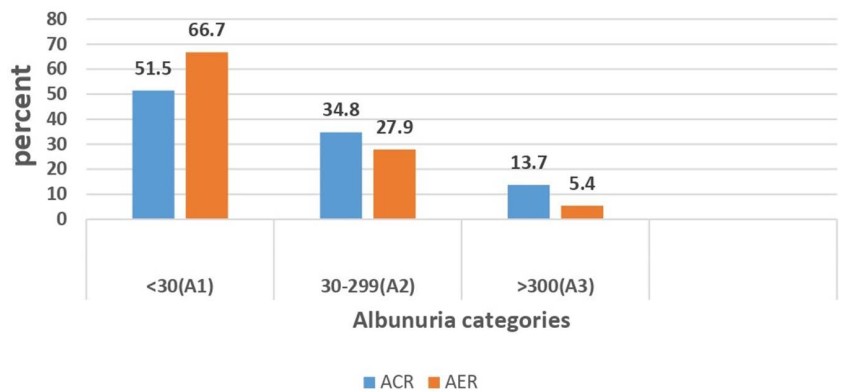

**Fig 1. Proportion of albuminuria using spot urine ACR (mg/g) and AER (mg/24hrs) in T2DM patients (N = 204), Gondar, Ethiopia, 2024.** ACR: Albumin to creatinine ratio (mg/g); AER: Albumin excretion rate (mg/24hrs); A1 = Normal; A2 = moderately elevated; A3 = severely elevated.

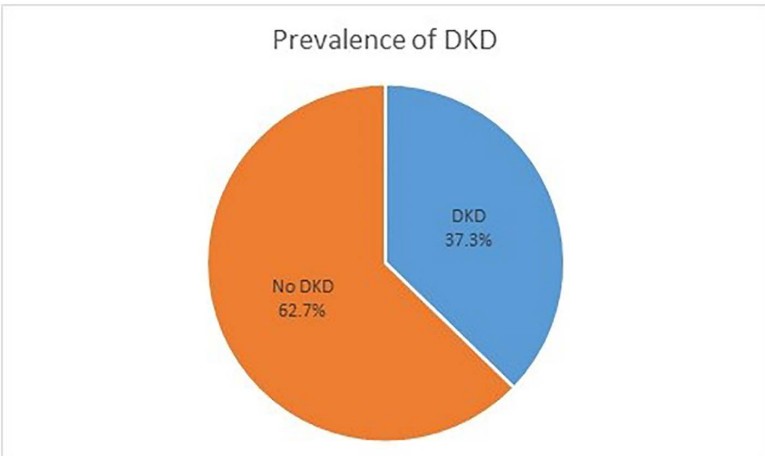

**Fig 2. Proportion of diabetic kidney disease (DKD) among patients with T2DM (N = 204), Gondar, Ethiopia, 2024.**

**Table 3. Proportion of DKD among T2DM patients using different diagnostic markers at the University of Gondar Specialized Hospital (N = 204), Gondar, Ethiopia, 2024.**

| Diagnostic marker | Number | Percentage |
|---|---|---|
| Spot urine ACR>=30 mg/g | 99 | 48.5 |
| 24 hr urine AER>=30 mg/24hrs | 68 | 33.3 |
| eGFR<60 mL/min/1.73m² using CKD-EPI creatinine equation | 13 | 6.4 |
| eGFR<60 mL/min/1.73m² using CKD-EPI creatinine and cystatin C equation | 15 | 7.4 |
| eGFR<60 mL/min/1.73m² using CKD-EPI cystatin C equation | 22 | 10.8 |

ACR: albumin to creatinine ratio; AER albumin excretion rate; eGFR: estimated glomerular filtration rate; CKD-EPI: Chronic Kidney Disease Epidemiology Collaboration.

As presented in Table 4, the multivariable analysis, found that place of residence, poor BP control, poor glycemic control, and duration of diabetes since diagnosis were significantly associated with the development of DKD. Urban participants had a 72.2% lower likelihood of developing DKD than those in rural areas (AOR = 0.278, 95% CI: 0.09–0.84, p = 0.023). Patients with poor BP control had over twice the risk of developing DKD compared to those with good BP control (AOR = 2.33, 95% CI: 1.17–4.66, p = 0.016), while he odds of having DKD were nearly three times higher in patients with poor glycemic control compared to those with good glycemic control (AOR = 2.93, 95% CI: 1.35–6.39, p = 0.007). Additionally, having diabetes for more than five years was significantly associated with a higher likelihood of developing DKD (AOR = 6.78, 95% CI: 2.93–15.70, p < 0.0001).

## Discussion

The prevalence of DKD, defined by a 24-hour AER ≥ 30 mg/24 hr or eGFRcr-cys < 60 mL/min/1.73 m², or both, was 37.3%. This result was more than double that of a previous report from the same study setting in Ethiopia, where the prevalence of DKD using eGFR (<60 mL/min/1.73 m²) alone was 17.3% [22]. The difference is likely due to variations in diagnostic criteria. While the previous study assessed DKD using only low eGFR, the current study incorporates both quantitatively measured albuminuria and eGFR determinations. Relying solely on eGFR decline as a screening tool for DKD can overlook cases where albuminuria is present but eGFR is still within normal range, as many patients exhibit albuminuria before experiencing a decrease in eGFR [9].

**Table 4. Factors associated with DKD in patients with T2DM at University of Gondar Specialized Hospital (N = 204), Gondar, Ethiopia, 2024.**

| Variable | Category | DKD | | COR(95% CI) | p value | AOR(95% CI) | p value |
|---|---|---|---|---|---|---|---|
| | | Yes | No | | | | |
| Residence | Urban | 65 | 119 | 0.45(0.18-1.13) | 0.09 | 0.278(0.09-0.84) | 0.023 |
| | Rural | 11 | 9 | 1 | | 1 | |
| Comorbidity | Yes | 55 | 73 | 1.97(1.07-3.64) | 0.03 | 1.70(0.86-3.38) | 0.129 |
| | No | 21 | 55 | 1 | | 1 | |
| BP control | Poor | 34 | 35 | 2.15(1.16-3.91) | 0.012 | 2.33(1.17-4.66) | 0.016 |
| | Good | 43 | 93 | 1 | | 1 | |
| Glycemic control | Poor | 63 | 84 | 2.54(1.26-5.11) | 0.009 | 2.93(1.35-6.39) | 0.007 |
| | Good | 13 | 44 | 1 | | 1 | |
| Duration of DM | >= 5 years | 66 | 73 | 4.97(2.35-10.54) | <0.0001 | 6.78(2.93-15.70) | <0.0001 |
| | < 5 years | 10 | 55 | 1 | | 1 | |

DKD: diabetic kidney disease; BP; blood pressure; DM: diabetes mellitus.

The current results reveal a higher prevalence than individual studies from Northeast Ethiopia (26.3%), Southwest Ethiopia (26%), Southern Ethiopia (18.2%), and Amhara region (10.8%), as well as a systematic review and meta-analysis report from Ethiopia (18.22%) [18–20,23]. However, it is similar to another systematic review and meta-analysis from Ethiopia, which reported a prevalence of 35.5% [17]. The lower proportions reported in previous studies may be due to differences in the markers used to assess DKD. Most studies relied on urine dipsticks and GFR estimations based on creatinine levels. Urine dipsticks can miss moderately elevated albuminuria and may underestimate the true prevalence. A positive urinary dipstick test is nearly always associated with an abnormal albumin-to-creatinine ratio (ACR). However, fewer than half of adults with both T2DM and an abnormal ACR have a positive dipstick test [25]. In contrast, our study utilized quantitatively measured albuminuria and incorporated cystatin C and creatinine-based equations; therefore, this study is likely to more accurately reflect the true prevalence of DKD.

The prevalence result from our study was comparable to studies from Nigeria (42.9%) and China (35.5%) but higher than the findings from a systematic review and meta-analysis of sub-Saharan Africa (29.7%) focusing on East Africa [16,28,29]. This discrepancy is likely related to differences in methodologies used to diagnose DKD.

The majority of patients were diagnosed to have DKD based on albuminuria criteria and only 7.4% participants have eGFR bellow 60 ml/min/1.732. This is due to the fact that patients with DKD can have albuminuria while remaining asymptomatic and maintaining a normal eGFR early in the diagnosis of DKD. In DKD, albuminuria often develops before a decline in estimated glomerular filtration rate (eGFR) because the initial pathogenic changes in the diabetic kidney primarily affect the glomerular filtration barrier rather than overall filtration function. Chronic hyperglycemia induces both structural and functional alterations in the glomerular capillary wall, including thickening of the basement membrane, podocyte injury or loss, widening of foot processes, and endothelial dysfunction with reduced nitric oxide production and decreased negative charge of the basement membrane. Proximal tubular dysfunction may also impair reabsorption of filtered albumin, contributing to albuminuria even when eGFR is still preserved. Hemodynamic changes, such as glomerular hyperfiltration, hyperperfusion, and elevated intraglomerular pressure, further promote early albumin leakage [30–32]. Lack of early screening and delayed diagnosis remain major challenges in low- and middle-income countries [33].

In the current study, 34.8% of patients with T2DM had moderately elevated albuminuria, while 13.7% had severely elevated albuminuria, as measured by ACR. This finding is consistent with studies from Egypt and a systematic review and meta-analysis on the prevalence of microalbuminuria in Africa [15,34].

Several factors have been associated with DKD in the literature. In the current study, place of residence, poor glycemic control, inadequate blood pressure control, and duration of diabetes mellitus since diagnosis were associated with DKD. Urban participants were less likely to develop DKD than their rural counterparts. This may be linked to better access to quality healthcare, higher education levels, and greater availability of information about diabetes. Rural populations with diabetes in low- and middle-income countries (LMICs) encounter significant challenges in meeting diabetes care performance metrics [35,36].

In the current study, poor glycemic control was associated with increased risk of developing DKD. Prior studies have also identified poor glycemic control as a major factor in the development and progression of DKD [17,21,37]. Elevated blood glucose levels contribute to kidney damage through mechanisms such as increased glomerular pressure and hyperfiltration, which can lead to glomerulosclerosis and the loss of nephrons. In this study, patients with uncontrolled hypertension had more than twice the risk of developing DKD, which aligns with previous research findings [37,38]. Additionally, a longer duration of diabetes was associated with higher DKD risk. Studies consistently show that the longer a patient has diabetes, the greater their risk of both developing and progressing to more severe stages of DKD [17,21,37–39]. Early and continuous management of blood glucose and blood pressure is critical in delaying the onset and slowing the progression of DKD in these patients.

Participants in this study were treated with metformin, sulfonylureas, and insulin. None of the participants were receiving SGLT2 inhibitors or GLP-1 receptor agonists. Emerging antidiabetic agents, particularly SGLT2 inhibitors and GLP-1 receptor agonists, have demonstrated renoprotective effects and are associated with reduced progression of chronic kidney disease [40,41]. Improving access to these medications in resource-limited settings is therefore essential.

## Strength and limitations of the study

This study has several important strengths. It is based on primary data collected specifically for the purpose of this investigation, in contrast to many previous studies from similar low- and middle-income settings that relied on secondary data or registry-based analyses. This approach allowed for standardized data collection and a more detailed and accurate assessment of diabetic kidney disease (DKD).

Another key strength is the comprehensive assessment of DKD using multiple validated markers. While many prior studies in similar settings defined DKD solely on the basis of estimated glomerular filtration rate (eGFR), thereby primarily identifying more advanced disease (CKD stage 3 and above), our study employed complementary measures, including spot urine albumin-to-creatinine ratio (ACR), 24-hour urine protein measurement (the gold standard for proteinuria assessment), and eGFR estimated using both serum creatinine and cystatin C. This multidimensional approach enabled the identification of early stages of DKD, including albuminuria in patients with preserved eGFR, which is particularly relevant for early detection and timely intervention.

Furthermore, this study represents one of the first detailed descriptions of DKD in an Ethiopian population with type 2 diabetes, contributing valuable data from a region where evidence remains scarce. The findings have important clinical and public health implications, particularly in low-resource settings where access to kidney replacement therapy is limited and early detection and prevention strategies are essential to reduce disease progression and associated morbidity.

Despite these strengths, several limitations should be acknowledged. First, the study was conducted in a tertiary care hospital, where patients with more complex or advanced type 2 diabetes mellitus are more likely to seek follow-up care. As a result, the findings may not be fully generalizable to diabetic patients managed in primary or community-based settings. Although DKD was defined clinically using albuminuria and reduced eGFR according to KDIGO guidelines, individuals with type 2 diabetes mellitus may have other causes of chronic kidney disease. Consequently, some patients without DKD may have been misclassified as having DKD. In addition, due to the cross-sectional nature of the study, some patients with transient albuminuria may have been incorrectly classified as having DKD.

Although both verbal and written instructions were provided for the 24-hour urine collection procedure, a 24-hour creatinine excretion test was not performed to verify collection completeness. As a result, errors in urine collection may have occurred, potentially affecting the accuracy of albuminuria measurements.

## Conclusion

The prevalence of DKD among patients with T2DM in this cohort, predominantly diagnosed through albuminuria, was high. Poor blood pressure control, inadequate glycemic control, and the duration of diabetes since diagnosis were significantly associated with the development of DKD. Regular screening for DKD, particularly using quantified albuminuria for early diagnosis, along with efforts to achieve good glycemic and blood pressure control, can help mitigate its progression.

## Supporting information

**S1 File. Completed STROBE checklist.**
(PDF)

**S2 File. Clinical and laboratory data.**
(XLS)

## Acknowledgments

We sincerely thank all the participants who generously contributed their time and information to this study. We also appreciate the dedication and efforts of the data collectors who supported the research process.

## Author contributions

**Conceptualization:** Workagegnehu Hailu, Tadesse Asmamaw Dejene, Markeshaw Tiruneh, Meseret Derbew Molla, Eshetie Melese Birru, Shitaye Alemu, Tadesse Awoke.

**Data curation:** Workagegnehu Hailu, Tadesse Asmamaw Dejene, Markeshaw Tiruneh, Meseret Derbew Molla, Eshetie Melese Birru, Tadesse Awoke.

**Formal analysis:** Workagegnehu Hailu, Tadesse Asmamaw Dejene, Markeshaw Tiruneh, Meseret Derbew Molla, Eshetie Melese Birru, Shitaye Alemu, Tadesse Awoke.

**Funding acquisition:** Workagegnehu Hailu, Tadesse Asmamaw Dejene, Markeshaw Tiruneh, Meseret Derbew Molla, Eshetie Melese Birru, Shitaye Alemu, Tadesse Awoke.

**Investigation:** Workagegnehu Hailu, Tadesse Asmamaw Dejene, Markeshaw Tiruneh, Meseret Derbew Molla, Eshetie Melese Birru, Shitaye Alemu, Tadesse Awoke.

**Methodology:** Workagegnehu Hailu, Tadesse Asmamaw Dejene, Markeshaw Tiruneh, Meseret Derbew Molla, Eshetie Melese Birru, Shitaye Alemu, Tadesse Awoke.

**Project administration:** Workagegnehu Hailu, Tadesse Asmamaw Dejene.

**Resources:** Workagegnehu Hailu, Tadesse Asmamaw Dejene, Markeshaw Tiruneh.

**Software:** Tadesse Asmamaw Dejene, Tadesse Awoke.

**Supervision:** Workagegnehu Hailu, Tadesse Asmamaw Dejene, Tadesse Awoke.

**Visualization:** Tadesse Asmamaw Dejene.

**Writing – original draft:** Workagegnehu Hailu.

**Writing – review & editing:** Workagegnehu Hailu, Tadesse Asmamaw Dejene, Markeshaw Tiruneh, Meseret Derbew Molla, Eshetie Melese Birru, Shitaye Alemu, Tadesse Awoke.

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
