## [Decision Letter · Decision Letter 0]

17 Nov 2025

Dear Dr. Hailu,

Thank you for submitting your manuscript to PLOS ONE. After careful consideration, we feel that it has merit but does not fully meet PLOS ONE’s publication criteria as it currently stands. Therefore, we invite you to submit a revised version of the manuscript that addresses the points raised during the review process.

**ACADEMIC EDITOR:**The paper sounds very interesting in addressing an emerging epidemic, the development of DKD, also in Africa. It is well-structured and clear-cut. Although the analysis is limited to a single experience it provides an important information in how DKD is also expanding in non western countries. As a main suggestion to appreciate the relevance of the study, it would be interesting to reinforce the strenghts of the study in the discussion. Why is this study informative despite its limitations? How does it compare to other studies conducted in similar countries? Is it larger? First describing DKD in a specific population? The descriptive nature of the study should be also better clarifies in the text. The Literature cited is quite limited and sometimes not updated. The Authors should follow the Reviewers suggestions to improve this section. Please pay particular attention to Reviewer #1 comments, which may improve the text. 

We look forward to receiving your revised manuscript.

Kind regards,

Francesca D'Addio, MD, PhD

Academic Editor

PLOS ONE

Journal Requirements:

“Funding for this research was obtained from the University of Gondar, College of Medicine and Health Sciences- research, technology transfer, and community engagement. (Ref No. R/T/T/C/Eng./189/11/22).”

“We are grateful to the research, technology transfer and community engagement office of the College of Medicine and Health Sciences, University of Gondar, for providing financial assistance for the study.”

“Funding for this research was obtained from the University of Gondar, College of Medicine and Health Sciences- research, technology transfer, and community engagement. (Ref No. R/T/T/C/Eng./189/11/22).”

4. We note that your Data Availability Statement is currently as follows: All relevant data are within the manuscript and its Supporting Information files

5. We notice that your supplementary figures are uploaded with the file type 'Other'. Please amend the file type to 'Supporting Information'. Please ensure that each Supporting Information file has a legend listed in the manuscript after the references list.

**Additional Editor Comments:**

The paper sounds very interesting in addressing an emerging epidemic, the development of DKD, also in Africa. It is well-structured and clear-cut. Although the analysis is limited to a single experience it provides an important information in how DKD is also expanding in non western countries. As a main suggestion to appreciate the relevance of the study, it would be interesting to reinforce the strenghts of the study in the discussion. Why is this study informative despite its limitations? How does it compare to other studies conducted in similar countries? Is it larger? First describing DKD in a specific population? The descriptive nature of the study should be also better clarifies in the text. The Literature cited is quite limited and sometimes not updated. The Authors should follow the Reviewers suggestions to improve this section. Please pay particular attention to Reviewer #1 comments, which may improve the text.

Reviewers' comments:

Reviewer's Responses to Questions

**Comments to the Author**

1. Is the manuscript technically sound, and do the data support the conclusions?

Reviewer #1: Partly

Reviewer #2: Yes

2. Has the statistical analysis been performed appropriately and rigorously?

Reviewer #1: I Don't Know

Reviewer #2: Yes

3. Have the authors made all data underlying the findings in their manuscript fully available?

Reviewer #1: Yes

Reviewer #2: Yes

4. Is the manuscript presented in an intelligible fashion and written in standard English?

Reviewer #1: Yes

Reviewer #2: Yes

Reviewer #1: 1) In the multivariable analysis, rural residence has an AOR < 1 (0.278), suggesting lower odds of DKD; however, the Discussion interprets rural residence as a risk factor. Can the authors confirm the reference category, re-check multivariable model coding and interpretation and correct text accordingly.

2) Since CKD-EPI and albuminuria categories were used, full KDIGO classification (G and A categories) should be reported. It would be good add a table summarizing G-stage and A-stage distributions.

3)Participants were recruited from a tertiary care center and may not reflect the community diabetic population, therefore it is necessary to clarify to what extent the cohort is representative.

4) Multiple statements imply causation, even if this is a cross-sectional study, I suggest to make a revision of the language using “associated with” rather than “predicts” or “leads to”

5) Can the authors add more details about the regression model, clarifying which variables were included and why they were chosen.

6) In the introduction I suggest to reduce the epidemioloy

7) Can the author clarify whether ACR measurement was repeated to confirm presence of albuminuria?

8) Can you clarify whether DKD prevalence differs between urban and rural subgroups?

9) In the discussion can the authors explain the mechanism behind early albuminuria with preserved eGFR?

10) The diabetic drugs in use are insulin, metformin, sulfonylureas which don’t provide diabetic kidney protection, can you make a comment regarding the neuw class of drugs such as SGLT2i and GLP-1RA mentioning the following articles:

-AWARE A novel web application to rapidly assess cardiovascular risk in type 2 diabetes mellitus Berra, C., Manfrini, R., Mirani, M., ... Fiorina, P., Folli, F. Acta Diabetologica, 2023, 60(9), pp. 1257–1266

-Improved glycemic and weight control with Dulaglutide addition in SGLT2 inhibitor treated obese type 2 diabetic patients at high cardiovascular risk in a real-world setting. The AWARE−2 study Berra, C., Manfrini, R., Bifari, F., ... Cimino, V., Folli, F. Pharmacological Research, 2024, 210, 107517

11) Check the english spelling and change Sulfonureas into Sufonylureas

Reviewer #2: The paper is overall well written and structured and give novel information concerning the prevalence of DKD in Ethiopia.

To put the paper in a bigger scenario, I suggest including a paragraph in the discussion concerning novel therapeutic approaches to couteract DKD. In particular the authors can consider to cite the following papers: PMID 35908663; PMID 36871895

**Do you want your identity to be public for this peer review?** For information about this choice, including consent withdrawal, please see our Privacy Policy

Reviewer #1: No

Reviewer #2: No

---

## [Author Response · Author response to Decision Letter 1]

15 Jan 2026

Answer to editor and Reviewers comment

Editor’s comment

The paper sounds very interesting in addressing an emerging epidemic, the development of DKD, also in Africa. It is well-structured and clear-cut. Although the analysis is limited to a single experience it provides an important information in how DKD is also expanding in non western countries. As a main suggestion to appreciate the relevance of the study, it would be interesting to reinforce the strenghts of the study in the discussion. Why is this study informative despite its limitations? How does it compare to other studies conducted in similar countries? Is it larger? First describing DKD in a specific population? The descriptive nature of the study should be also better clarifies in the text. The Literature cited is quite limited and sometimes not updated. The Authors should follow the Reviewers suggestions to improve this section. Please pay particular attention to Reviewer #1 comments, which may improve the text.

Response to editor’s comment

Thank you for reviewing our manuscript and for your constructive and insightful feedback. We have carefully addressed your comments as well as those of the reviewers.

1. In response to your suggestion to better emphasize the relevance of the study, we have revised the Discussion section by renaming the final subsection from “Limitations” to “Strengths and Limitations” and by explicitly highlighting the strengths of our research.

The key strength of our study is that it is based on primary data collected specifically for this investigation, whereas many previous studies from similar settings relied on secondary or registry data. In addition, several prior studies defined DKD solely using eGFR, which primarily captures more advanced disease (CKD stage 3 and above). In contrast, our study employed multiple complementary markers of DKD, including spot urine albumin-to-creatinine ratio (ACR), 24-hour urine protein measurement, and eGFR estimated using both creatinine and cystatin C. This comprehensive approach allowed us to identify early stages of DKD, including albuminuria in patients with preserved eGFR, which is particularly relevant for early detection and prevention strategies in resource limited settings where access to kidney replacement therapy is limited.

See “Strengths and Limitation” section

2. Descriptive nature of the study- Thank you for this comment. We would like to clarify that our study is an analytical cross-sectional study with a descriptive component, designed to estimate the prevalence of diabetic kidney disease and to examine factors associated with its presence.

3. Thank you for your comment regarding the literature cited. We have carefully reviewed the reference list and updated it to include more recent and relevant studies on diabetic kidney disease, particularly from low- and middle-income countries. Additional references have been added throughout the manuscript to provide a more comprehensive and current context for our study.

Response to Reviewer #1 comment

Comment 1: In the multivariable analysis, rural residence has an AOR < 1 (0.278), suggesting lower odds of DKD; however, the Discussion interprets rural residence as a risk factor. Can the authors confirm the reference category, re-check multivariable model coding and interpretation and correct text accordingly.

Response: Thank you for this comment. We have re-checked the coding of the multivariable model and confirmed that urban residence is the exposure with an AOR < 1 (AOR = 0.278), indicating a protective effect compared with rural residence (the reference category). The Results and Discussion have been reviewed and are consistent with this interpretation. (See Table 4 and Discussion section)

Comment 2: Since CKD-EPI and albuminuria categories were used, full KDIGO classification (G and A categories) should be reported. It would be good add a table summarizing G-stage and A-stage distributions.

Response: Thank you for this important suggestion. In our study, the majority of DKD cases were identified based on albuminuria, and the albuminuria categories (A-stages) are presented in Figure 1. In addition, the proportion of DKD diagnosed using different criteria is summarized in Table 3. As shown in Table 3, only 7.4% of participants met the CKD definition based on reduced eGFR (CKD-EPI creatinine, eGFR < 60 mL/min/1.73 m²).

We explored presenting the full KDIGO G and A combined staging (e.g., G1A2, G1A3, G2A2, G2A3, G3A1, G3A2, G3A3 etc). However, because the proportion of participants with G3 or higher stages was very small, this resulted in a lengthy table with several cells containing very low frequencies, which we felt would add limited interpretive value. Therefore, we opted to present the data as currently shown.

Nevertheless, we are happy to include a table summarizing the full KDIGO G and A staging if the reviewer feels it is necessary.

Comment 3: Participants were recruited from a tertiary care center and may not reflect the community diabetic population, therefore it is necessary to clarify to what extent the cohort is representative.

Response: Thank you for this valid comment. We have addressed this issue by adding the following statement to the Limitations section to clarify the representativeness of the study population:

“The study was conducted in a tertiary care hospital, where patients with more complex or advanced type 2 diabetes mellitus are more likely to seek follow-up care. As a result, the findings may not be fully generalizable to diabetic patients managed in primary or community-based settings.”

Comment 4: Multiple statements imply causation, even if this is a cross-sectional study, I suggest to make a revision of the language using “associated with” rather than “predicts” or “leads to”

Response: Thank you for this important comment. We have revised the manuscript to remove causal language and replaced by “associated with”

Comment 5: Can the authors add more details about the regression model, clarifying which variables were included and why they were chosen.

Response: Thank you for this comment. We have clarified the regression modeling approach in the Methods section (under “statistical analysis”) and Results section (under “Factors associated with DKD in patients with T2DM.” Variables with a p-value ≤ 0.20 in the bivariate analysis—namely place of residence, presence of comorbidities, blood pressure control status, glycemic control status, and duration of diabetes—were selected for inclusion in the multivariable logistic regression model.

Comment 6: In the introduction I suggest to reduce the epidemiology

Response: Thank you for this suggestion. We have revised the Introduction to reduce the epidemiology content and improve focus and conciseness.

Comment 7: Can the author clarify whether ACR measurement was repeated to confirm presence of albuminuria?

Response: Thank you for this important question. The spot urine albumin-to-creatinine ratio (ACR) measurement was not repeated. However, all participants additionally underwent quantitative urine protein assessment using a 24-hour urine protein excretion test, which was used as the reference (gold standard) to confirm the presence of albuminuria and to define DKD in this study. To clarify this we put the following in the methods (under blood and urine collection and processing section) “A first-morning midstream urine sample was collected from all participants for measurement of the spot urine albumin-to-creatinine ratio (ACR). The ACR measurement was performed once and was not repeated. In addition, all participants underwent quantitative assessment of albuminuria using a 24-hour urine protein excretion test”

Comment 8: Can you clarify whether DKD prevalence differs between urban and rural subgroups?

Response: Thank you for this question. DKD prevalence was 35.3% in urban participants (65/184) and 55.0% in rural participants (11/20). A chi-square test showed no statistically significant difference between the groups (χ² = 2.20, p = 0.14). In the multivariable analysis, urban residence was associated with lower odds of DKD (AOR = 0.278, Table 4), as discussed in the manuscript.

Comment 9: In the discussion can the authors explain the mechanism behind early albuminuria with preserved eGFR?

Response: Thank you for this valid comment. We have addressed this by adding a dedicated paragraph in the Discussion section explaining the pathophysiological mechanisms underlying the development of albuminuria in the presence of preserved eGFR, within the section where albuminuria is discussed. We have added the following paragraph with respective references

“In DKD, albuminuria often develops before a decline in estimated glomerular filtration rate (eGFR) because the initial pathogenic changes in the diabetic kidney primarily affect the glomerular filtration barrier rather than overall filtration function. Chronic hyperglycemia induces both structural and functional alterations in the glomerular capillary wall, including thickening of the basement membrane, podocyte injury or loss, widening of foot processes, and endothelial dysfunction with reduced nitric oxide production and decreased negative charge of the basement membrane. Proximal tubular dysfunction may also impair reabsorption of filtered albumin, contributing to albuminuria even when eGFR is still preserved. Hemodynamic changes, such as glomerular hyperfiltration, hyperperfusion, and elevated intraglomerular pressure, further promote early albumin leakage”.

Comment 10: The diabetic drugs in use are insulin, metformin, sulfonylureas which don’t provide diabetic kidney protection, can you make a comment regarding the neuw class of drugs such as SGLT2i and GLP-1RA mentioning the following articles:

-AWARE A novel web application to rapidly assess cardiovascular risk in type 2 diabetes mellitus Berra, C., Manfrini, R., Mirani, M., ... Fiorina, P., Folli, F. Acta Diabetologica, 2023, 60(9), pp. 1257–1266

-Improved glycemic and weight control with Dulaglutide addition in SGLT2 inhibitor treated obese type 2 diabetic patients at high cardiovascular risk in a real-world setting. The AWARE−2 study Berra, C., Manfrini, R., Bifari, F., ... Cimino, V., Folli, F. Pharmacological Research, 2024, 210, 107517

Response: Thank you for this valuable suggestion. We have added a paragraph in the Discussion highlighting the potential of novel therapeutic approaches, including SGLT2 inhibitors and GLP-1 receptor agonists, in slowing the progression of diabetic kidney disease. We also emphasize the importance of improving access to and appropriate use of these effective therapies in resource-limited settings like Ethiopia. We have included citation related to this.

Comment 11: Check the english spelling and change Sulfonureas into Sufonylureas

Response: Thank you for noting this. We have corrected the spelling accordingly

Response to Reviewer #2

Comment 1: The paper is overall well written and structured and give novel information concerning the prevalence of DKD in Ethiopia.

Response: Thank you for your positive comment and appreciation of our work

Comment 2: To put the paper in a bigger scenario, I suggest including a paragraph in the discussion concerning novel therapeutic approaches to couteract DKD. In particular the authors can consider to cite the following papers: PMID 35908663; PMID 36871895

Response: Thank you for this valuable suggestion. We have added a paragraph in the Discussion highlighting the potential of novel therapeutic approaches, including SGLT2 inhibitors and GLP-1 receptor agonists, in slowing the progression of diabetic kidney disease. We also emphasize the importance of improving access to and appropriate use of these effective therapies in resource-limited settings like Ethiopia. The suggested references have been cited accordingly

---

## [Decision Letter · Decision Letter 1]

4 Feb 2026

Diabetic Kidney Disease in Northwest Ethiopia: Prevalence and Determinants Among Adults with Type 2 Diabetes

PONE-D-25-53996R1

Dear Dr. Hailu,

We’re pleased to inform you that your manuscript has been judged scientifically suitable for publication and will be formally accepted for publication once it meets all outstanding technical requirements.

Kind regards,

Francesca D'Addio, MD, PhD

Academic Editor

PLOS One

Additional Editor Comments (optional):

By responding for the most part to the main comments raised by the Reviewers, the paper has been significantly improved and it is now suitable for publication.

Reviewers' comments:

Reviewer's Responses to Questions

**Comments to the Author**

Reviewer #1: All comments have been addressed

Reviewer #2: All comments have been addressed

2. Is the manuscript technically sound, and do the data support the conclusions?

Reviewer #1: Yes

Reviewer #2: Yes

3. Has the statistical analysis been performed appropriately and rigorously?

Reviewer #1: Yes

Reviewer #2: Yes

4. Have the authors made all data underlying the findings in their manuscript fully available?

Reviewer #1: Yes

Reviewer #2: Yes

5. Is the manuscript presented in an intelligible fashion and written in standard English?

Reviewer #1: Yes

Reviewer #2: Yes

Reviewer #1: The authors has adequately addressed the comments and now the article has reached a good level and It is now acceptable for publication, it is presented in a n intelligible fashion and written in standard english.

Reviewer #2: The authors addressed all reviewers' comments. I just would like to highlight that the bibliography han not been updated as compared to the previous version.

**Do you want your identity to be public for this peer review?** For information about this choice, including consent withdrawal, please see our Privacy Policy

Reviewer #1: No

Reviewer #2: No

---

## [Editor Report · Acceptance letter]

PONE-D-25-53996R1

PLOS One

Dear Dr. Hailu,

I'm pleased to inform you that your manuscript has been deemed suitable for publication in PLOS One. Congratulations! Your manuscript is now being handed over to our production team.

Kind regards,

on behalf of

Prof. Francesca D'Addio

Academic Editor

PLOS One